# FLOWING 🌊: Implicit Neural Flows
# for Structure-Preserving Morphing

**Arthur Bizzi**[1]    **Matias Grynberg**[2]    **Vitor Matias**[3]    **Daniel Perazzo**[4,5]    **João Paulo Lima**[4,6]

**Luiz Velho**[4]    **Nuno Gonçalves**[7,8]    **João Pereira**[9]    **Guilherme Schardong**[7]    **Tiago Novello**[4]

[1]EPFL    [2]University of Buenos Aires    [3]University of São Paulo    [4]IMPA    [5]CSML-IIT
[6]Universidade Federal Rural de Pernambuco    [7]ISR-UC    [8]INCM    [9]University of Georgia

## Abstract

Morphing is a long-standing problem in vision and computer graphics, requiring a time-dependent warping for feature alignment and a blending for smooth interpolation. Recently, multilayer perceptrons (MLPs) have been explored as implicit neural representations (INRs) for modeling such deformations, due to their meshlessness and differentiability; however, extracting coherent and accurate morphings from standard MLPs typically relies on costly regularizations, which often lead to unstable training and prevent effective feature alignment. To overcome these limitations, we propose FLOWING (**FLOW** morph**ING**), a framework that recasts warping as the construction of a differential vector flow, naturally ensuring continuity, invertibility, and temporal coherence by encoding structural flow properties directly into the network architectures. This flow-centric approach yields principled and stable transformations, enabling accurate and structure-preserving morphing of both 2D images and 3D shapes. Extensive experiments across a range of applications—including face and image morphing, as well as Gaussian Splatting morphing—show that FLOWING achieves state-of-the-art morphing quality with faster convergence. Code and pretrained models are available in
`https://schardong.github.io/flowing`.

## 1  Introduction

Morphing is a long-standing problem in computer vision and graphics [12; 56], with applications in image editing [48; 4], biometrics [46; 13], and 3D shape interpolation [27; 31; 42]. The task consists of continuously interpolating between two signals while ensuring that the intermediate representations remain structurally consistent. Traditionally, morphing is decomposed into two stages: a **warping** stage, which aligns source and target features over time (ideally through a one-parameter family of diffeomorphisms), and a **blending** stage, which interpolates between the aligned signals.

Recently, ifmorph [44] introduced a face morphing approach that employs *multilayer perceptrons* (MLPs) as *implicit neural representations* (INRs) to model the warping transformation. INRs provide a differentiable and memory-efficient solution for representing low-dimensional signals and have been successfully applied to surface reconstruction [47; 14; 58; 30; 45], surface evolution [27; 31; 41], radiance fields [28; 3], and image modeling [1; 34; 35]. However, applying generic MLPs to represent warping transformations presents important limitations. For instance, enforcing temporal coherence requires explicit regularization in the loss function, which greatly increases training time and may lead to convergence issues. In addition, unconstrained MLPs lack structural priors, making the learned deformation prone to undesirable behaviors such as singularities—regions where the mapping becomes non-invertible—resulting in artifacts that degrade the morphing quality (see the first row in Figure 1, bottom-left).

39th Conference on Neural Information Processing Systems (NeurIPS 2025).

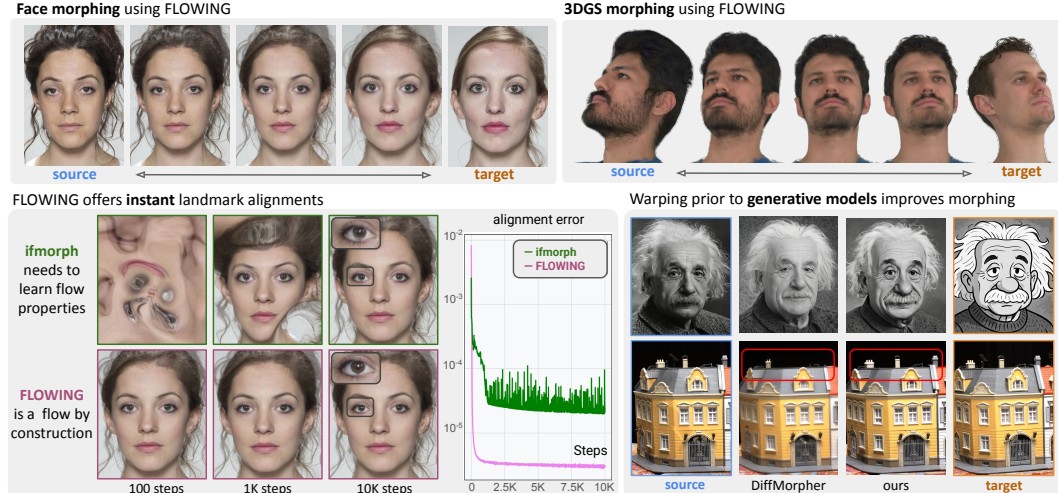

Figure 1: We present **FLOWING**, a robust and theoretically grounded framework for fast, accurate, and structure-preserving morphing. It enables smooth and temporally consistent morphs by learning a structure-preserving flow, applicable to both 2D (top-left) and 3D (top-right) data; the latter uses 3DGS for representing 3D faces. Bottom-left: FLOWING instantly aligns landmarks by construction, outperforming SoTA methods in both visual quality and convergence speed. Bottom-right: Integrating FLOWING with generative models improves morphing fidelity and semantic coherence.

To address these limitations, we propose **FLOWING**, a framework that employs specialized flow-based INRs that perform **structure-preserving warpings by construction**. We reframe morphing as the problem of constructing an interpolating flow, allowing us to inherit the structural properties of flow operators through flow-based neural architectures, such as *neural ODEs* (NODEs) [9] and *neural conjugate flows* (NCFs) [6]. FLOWING adapts these architectures to the 2D/3D warping contexts and leverages their architectural priors to produce morphings that are valid by construction. As a result, it enables near-instant training, significantly faster than generic MLPs that depend on soft regularization to enforce coherence. Figure 1 showcases FLOWING's ability to generate smooth, consistent, and structure-preserving morphs in both 2D images and *3D Gaussian splatting* (3DGS) [20], with clear improvements in visual quality, alignment, speed, and semantic coherence over prior methods. In summary, our contributions are:

- A principled, flow-based morphing approach that leverages specialized INRs to enforce continuity and temporal coherence by construction. This enables learning highly detailed, structure-preserving morphings near instantly, using only sparse landmark correspondences.

- An adaptation of flow-based architectures to the morphing setting, combined with SIRENs [47; 32], allowing the representation of complex, high-frequency deformations and accurate keypoint matching. To minimize spurious deformations, we incorporate thin-plate energy regularization [7; 49], enabled by a novel forward-mode differentiation scheme.

- Demonstration of our approach across diverse morphing tasks—including face morphing, general image morphing, and 3DGS morphing—achieving strong results in 3D splat morphing via a novel blending scheme.

## 2   Related works

**Warping and morphing** techniques have been widely studied in computer graphics and vision [12; 56; 55]. Classical approaches rely on geometric transformations such as thin-plate splines [26], radial basis functions [7], and triangle meshes to morph between images [4; 49]. While effective, these methods often struggle to handle smooth and complex deformations. Moreover, standard OpenCV-style morphing combines simple linear interpolation between features with a shared triangulated domain, which is not always available. In contrast, by introducing flow concepts into the model architecture, our method (FLOWING) achieves high-quality warping for both images and 3D data. It

requires only feature correspondences as input and supports nonlinear interpolation between features, enabling more flexible and robust morphing.

**Implicit neural representations (INRs)** have gained significant attention for encoding continuous low-dimensional signals directly in the parameters of neural networks. Early works such as SIREN [47] and Fourier feature networks [52] enhanced INR expressiveness by mapping input coordinates to sinusoidal functions, enabling detailed reconstructions of images, 3D shapes, and physical fields. More recently, INRs have been applied to 2D face morphing [44], where SIRENs are used to parameterize the warping and additional costly constraints are introduced to enforce flow properties. In contrast, our method leverages flow-based architectures that enable faster, more efficient, and robust morphing of both 2D and 3D data, with deformations learned as continuous, structure-preserving flows.

**Flow-based networks** have been explored for modeling continuous-time dynamics and transformations. NODEs [9] provide a powerful yet computationally intensive framework for learning continuous-time dynamical systems, where inference through discrete layers is replaced by numerical integration. Originally developed for continuous normalizing flows, NODEs have since been applied to medical image registration [2; 57; 50; 51], solving PDEs [5]. More recently, NCFs [6] were introduced as an alternative approach to model dynamics by topologically deforming affine flows, enabling greater efficiency through parallelism. In this work, we adapt both NODEs and NCFs to the morphing setting for images and 3DGS, further enhancing their representational power with sinusoidal activations.

**Generative methods** have also emerged as a powerful and flexible approach for morphing, with DiffMorpher [59] presenting general image morphing without reference landmarks. However, they face two key limitations: their outputs are constrained to a fixed resolution, and they require pre-aligned target images, adding significant preprocessing overhead. Our method addresses these challenges by introducing flow-based warping, which acts as a sophisticated, non-linear alignment mechanism. This enables seamless integration with generative blending techniques, improving both alignment quality and applicability.

## 3 FLOWING

### 3.1 Problem setup: morphing as flows

Given source and target media objects $f^0, f^1 : \Omega \subset \mathbb{R}^n \to \mathbb{R}^m$, where $\Omega$ denotes their supports, annotated with $K$ feature correspondences $\{p_i^0, p_i^1\} \subset \Omega$, our goal is to construct a continuous, time-dependent family of warpings $\Phi : \Omega \times [0, 1] \to \mathbb{R}^n$ that allows us to smoothly interpolate between the source features at $t = 0$ and the corresponding target features at $t = 1$, such that $\Phi(p_i^0, 1) = p_i^1$. For morphing, $\Phi$ should have additional properties:

• **Uniqueness.** The path traced by each feature must be unique, otherwise distinct features may overlap, leading to unstructured, incoherent interpolations.

• **Invertibility.** The forward and backward mappings of source and target features should match at intermediate times and remain symmetric under time reversal to avoid artifacts during blending.

• **Energy-minimization.** Feature paths should be minimal and smooth for coherent interpolation; otherwise, overfitting may introduce artifacts or singularities that degrade morphing quality.

The key idea behind FLOWING is that the first two properties arise from the definition of **flows**: by constraining features to move as an integral over an underlying vector field, we may leverage the uniqueness and reversibility of its orbits. This allows us to recast the morphing problem as the construction of a flow operator [54], enforcing corresponding features to remain consistent over time:

$$\Phi(p_i^0, t) = \Phi(p_i^1, t - 1) \quad \text{for } i \in \{1, \dots, K\} \text{ and } t \in [0, 1]. \tag{1}$$

To enforce these structural properties on the warping map, we employ a $\theta$-parametrized flow representation $\Phi_\theta$ (see Sec. 3.3). The third property is imposed by minimizing the curvature of the orbits of $\Phi_\theta$ across the domain $\Omega$. This is achieved with the following thin-plate-like optimization problem:

$$\arg\min_\theta \|\mathbf{J}\Phi_\theta'(\mathbf{x}, 0)\|_2^2 \quad \text{subject to} \quad \Phi_\theta(p_i^0, t) = \Phi_\theta(p_i^1, t - 1), \tag{2}$$

where $\Phi_\theta'(\mathbf{x}, 0)$ denotes the time derivative of $\Phi_\theta(\mathbf{x}, t)$ at $t = 0$ and $\mathbf{J}\Phi_\theta'$ its Jacobian. Once the features $\{p_i^0, p_i^1\}$ are aligned, the resulting flow can be used to warp the source and target signals to

an intermediate time, after which the resulting warpings are blended together. Figure 2 provides an overview of the morphing process.

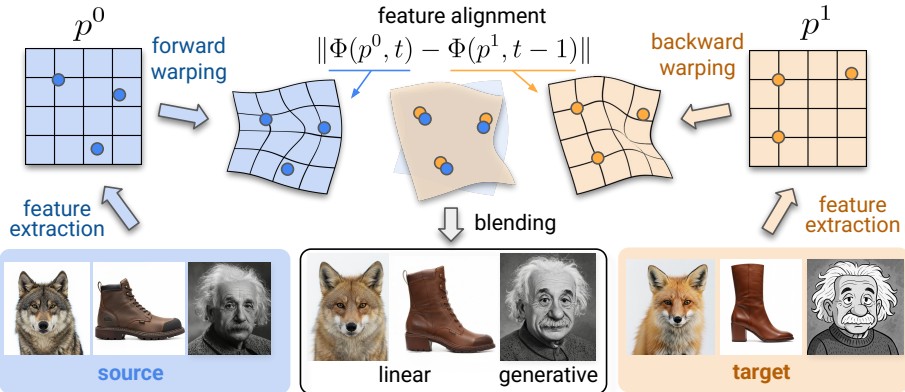

Figure 2: **Overview of FLOWING.** Given source and target images $I^0, I^1$ we extract landmark pairs $(p_i^0, p_i^1)$ with a feature extractor (`dlib`, Xfeat, etc). We train a flow $\phi$ such that $\phi(p^0, t) = \phi(p^1, t-1)$, effectively mapping $p^0$ to $p^1$. At inference, we warp $I^0$ forward by $t$ units and $I^1$ and backward by $t - 1$ units, then blend them together with methods such as linear blending or generative models.

## 3.2 Training

Training a flow-based architecture $\Phi_\theta$ requires defining a loss function to solve the optimization problem in (2). Note that the uniqueness and invertibility properties are guaranteed by construction, since $\Phi_\theta$ is a flow. Therefore, the loss function $\mathcal{L}$ consists of a data constraint to enforce feature matching and a regularization term to penalize path distortion:

$$\mathcal{L}(\theta) = \underbrace{\sum_i \int_{[0,1]} \left\| \Phi_\theta(p_i^0, t) - \Phi_\theta(p_i^1, t-1) \right\|_2^2 \, dt}_{\text{Data constraint}} + \lambda \underbrace{\int_\Omega \left\| \mathbf{J}\Phi_\theta'(\mathbf{x}, 0) \right\|_2^2 \, d\mathbf{x}}_{\text{Thin-plate constraint}}, \qquad (3)$$

where $\lambda > 0$ is a parameter. In practice, the data term is enforced on only a few time steps, since uniqueness ensures that feature correspondences hold across the interval. We find out that using $t = 0.5$ for NODE and $t = \{0, 0.25, 0.5, 0.75, 1\}$ for NCF is sufficient.

For thin-plate term, we minimize the Jacobian of $\Phi_\theta'(\mathbf{x}, 0)$ over the domain, which is equivalent to minimizing the second derivative of the integral path $\Phi(\mathbf{x}, t)$ for each $\mathbf{x}$. Since $\Phi$ is a flow, explicit time sampling is unnecessary—one reason flow-based architectures train faster than generic MLPs. The integral is approximated via Monte Carlo methods, with $\lambda$ as the regularization weight. To further accelerate computation, we implement a *forward differentiation* (FD) scheme based on generalized dual-number arithmetic, enabling derivatives to be computed in parallel with inference, significantly reducing overhead. Our ablation study shows that FD makes the thin-plate loss calculation **23 times faster** than standard `autograd`. Full results and details are given in Appendix A.

## 3.3 Flow-based architectures

Flow-based networks provide a principled representation for the time-dependent warping $\Phi_\theta$, as they hold desirable properties such as continuity, invertibility, and temporal coherence. Compared to using MLPs to parametrize the flow operator $\Phi_\theta$, which require expensive additional terms to approximate uniqueness and invertibility in addition to (2), flow-based architectures enforce these properties by construction. As a result, they rely on simpler losses, may achieve near-instant training, and provide accurate alignment while avoiding the catastrophic artifacts that arise when MLPs fail to capture proper flow properties.

Figure 3 compares an MLP-based flow (ifmorph [44]) with our approach (FLOWING). For this experiment, we employed `dlib` to extract features centered on the faces, which explains why the ears are not aligned. In this work, we focus on two flow-based networks: Neural ODEs and Neural Conjugate Flows.

**Neural ODEs.** First, we propose using NODEs [9] to model the vector field $\Phi'$, the time derivative of the warping deformation $\Phi$, with a neural network $\mathcal{F}_\theta$. Thus, inference for these models involves performing numerical integration over the vector field $\mathcal{F}_\theta$; analogously, it may be interpreted as deep ResNets[15] with "continuous depth". Specifically, NODEs take the following formulation:

$$\frac{d}{dt}\mathbf{x} = \mathcal{F}_\theta(\mathbf{x}) \implies$$
$$\Phi(\mathbf{x}_0, t) = \mathbf{x}(0) + \int_0^t \mathcal{F}_\theta(\mathbf{x}(\tau))d\tau. \tag{4}$$

To achieve high-quality matching, the vector field $\mathcal{F}_\theta$ is modeled as a SIREN [47], allowing it to capture highly detailed spatial deformations necessary to handle the diverse range of distances and orbits traversed by each pair of correspondences. Moreover, to ensure that orbit deformations remain minimal, we penalize the norm of the Jacobian matrix of $\mathcal{F}_\theta$ over the domain $\Omega$.

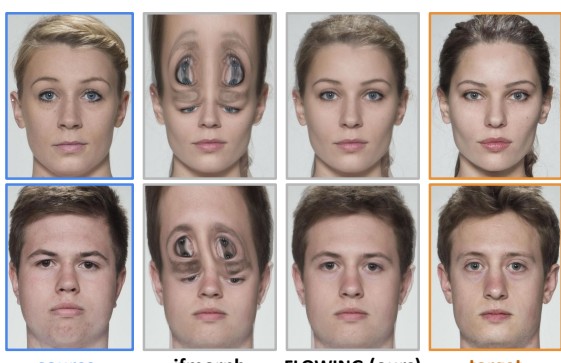

source        ifmorph        FLOWING (ours)        target

Figure 3: Comparison between ifmorph and FLOWING. While ifmorph fails to preserve structure, FLOWING produces clean and realistic interpolations by enforcing flow properties such as invertibility and trajectory uniqueness.

During training, we integrate $\mathcal{F}_\theta$ using a fourth-order Runge–Kutta method. We integrate the source and target points to $t = 0.5$ using the same number of integration steps for the forward and backward dynamics. Since NODE-based approaches are often prone to numerical errors during integration, we include in Appendix B an ablation study on the total number of integration steps involved in this process, showing that we can use very few integration steps while both preserving an accurate approximation of the vector field $\mathcal{F}_\theta$ and keeping the training efficient. Moreover, at inference time, the number of integration steps can be increased arbitrarily, without additional training cost, to more accurately capture the dynamics of $\mathcal{F}_\theta$.

**Neural conjugate flows.** While NODEs explicitly integrate neural vector fields, they can become computationally expensive when many steps are required during inference. Neural conjugate flows (NCFs) [6] offer an alternative formulation based on topological conjugation. Instead of sequential integration, NCFs employ invertible neural networks to deform the orbits of simple affine flows, enforcing a simplified topology. Precisely, the flow-based warping is given by:

$$\Phi_\theta(\mathbf{x}, t) = H_\theta^{-1} \circ \Psi\big(H_\theta(\mathbf{x}), t\big), \quad (\mathbf{x}, t) \in \mathbb{R}^n \times \mathbb{R}. \tag{5}$$

We parameterize $H_\theta$ and $H_\theta^{-1}$ using invertible architectures known as coupling layers [11]. These apply alternating, memory-equipped transformations in sequence, guaranteeing invertibility by construction. Importantly, the resulting conjugated flow $\Phi_\theta$ remains a valid flow, inheriting key properties such as continuity, invertibility, and associativity from the original affine system.

### 3.4 Blending

Having established how FLOWING constructs flow-based warpings that guarantee structural properties and accurate feature alignment, we now turn to the second stage of morphing: **blending**.

**Image morphing.** Let $I^0$, $I^1 : \mathbb{R}^2 \to \mathcal{C}$ denote two input images and $\Phi_\theta : \mathbb{R}^2 \times \mathbb{R} \to \mathbb{R}^2$ be a flow aligning their features over time. We define the warped images as $\mathcal{I}^i(\mathbf{x}, t) = I^i\big(\Phi_\theta(\mathbf{x}, i - t)\big)$, ensuring spatial alignment at each intermediate timestep $t \in [0, 1]$. A straightforward morphing is then obtained by linearly blending the aligned images:

$$\mathcal{I}(\mathbf{x}, t) = (1 - t)\mathcal{I}^0(\mathbf{x}, t) + t\mathcal{I}^1(\mathbf{x}, t), \tag{6}$$

yielding a smooth function $\mathcal{I} : \mathbb{R}^2 \times \mathbb{R} \to \mathcal{C}$ that interpolates between $I^0$ and $I^1$.

While generative models can interpolate between images, they typically lack explicit feature alignment. To address this limitation, we first apply our flow-based warping $\Phi_\theta$ to align features over time, and

then perform blending in the latent space of a pretrained generative model. Let $\mathcal{E}$ and $\mathcal{D}$ denote the encoder and decoder of a generative model. We define generative morphing as:

$$\mathcal{I}(\cdot, t) = \mathcal{D}\left((1-t)c^0(t) + tc^1(t)\right), \quad \text{where} \quad c^i(t) = \mathcal{E}\left(\mathcal{I}^i(\cdot, t)\right). \tag{7}$$

This strategy combines explicit spatial alignment with perceptual quality, producing temporally coherent transitions that significantly outperform naive latent-space blending.

**3D Gaussian Splatting morphing.** Beyond images, we show that morphing with FLOWING can be extended to 3DGS [20; 36] which has recently emerged as a popular format for 3D representation. 3DGS enables photorealistic rendering from a set of Gaussians $\mathcal{G}$ with each Gaussian $g_k = \{p_k, \alpha_k, c_k, \Sigma_k\}$ consisting of a center $p_k \in \mathbb{R}^3$, opacity $\alpha_k$, view-dependent color $c_k \in \mathbb{R}^3$, and covariance matrix $\Sigma_k \in \mathbb{R}^{3\times3}$ that encodes orientation and spread.

Our method morphs between two Gaussian sets, $\mathcal{G}^0$ and $\mathcal{G}^1$, producing an intermediate set $\mathcal{G}(t)$ with $\mathcal{G}(0) = \mathcal{G}^0$ and $\mathcal{G}(1) = \mathcal{G}^1$. We employ a FLOWING network $\Phi_\theta$ to smoothly align Gaussian centers over time, while linearly blending the opacity parameters $\alpha_k^i$ to ensure seamless transitions. Specifically, for each Gaussian $g_k^i \in \mathcal{G}^i$, we compute:

$$g_k^i(t) = \left\{\Phi_\theta(p_k^i, \ t-i), \ |1-i-t| \alpha_k^i, \ c_k^i, \ \Sigma_k^i\right\}, \tag{8}$$

which jointly applies warping and blending. The final morphed set is then defined by $\mathcal{G}(t) = \mathcal{G}^0(t) \cup \mathcal{G}^1(t)$. This 3DGS morphing procedure preserves structural coherence through flow-based alignment and achieves photorealistic 3D transitions via our linear blending strategy.

## 4  Experiments

We evaluate FLOWING on face and image morphing, as well as on 3DGS morphing, using four diverse datasets. These experiments demonstrate the effectiveness of our approach in both 2D and 3D settings. Additionally, we provide ablation studies in Appendix B to validate our architectural choices and regularization strategies.

**Evaluation datasets.** For face images, we use the FRLL dataset [10], which contains 102 identities with 5 images captured at fixed angles and two expressions (neutral and smiling). We use the neutral, front-facing images. Following the protocol in [43], we construct 1220 pairs, with landmarks extracted using `dlib` [19; 40]. As a post-processing step, we apply FFHQ alignment and cropping, producing images at a resolution of $1350^2$. The second dataset is a subset of MegaDepth [25], which provides multi-view scene landmarks. From this subset, we extract 275 image pairs based on the cosine similarity between their ResNet [15] embeddings, with feature correspondences obtained using Xfeat [38]. Third, we use eight subjects from NeRSemble [22], a multi-view collection of human heads. Following [39], we fit an FLAME model to each subject to extract 254 landmarks per head. Finally, we include in-the-wild face images from FFHQ [18] for qualitative evaluation.

FLOWING is implemented in PyTorch [33] and trained with the Adam optimizer [21]. At each training step, we sample 20,000 points from the spatial domain $[-1, 1]^2$ for the selected values of $t$.

### 4.1  Quantitative comparisons

To evaluate the performance of FLOWING on the morphing task, we consider two key components: **warping**, which aligns landmarks across signals over time, and **blending**, which generates smooth intermediate transitions. Accordingly, we organize our comparisons into two aspects: *landmark alignment* (4.1.1) and *blending quality* (4.1.2).

#### 4.1.1  Landmark alignment

We compare FLOWING with ifmorph [44], testing both NCF and NODE backbones with sigmoid and SIREN activations. For a fair comparison, all methods are supervised under the same set of landmark points.

**Metrics.** We report the *mean squared error* (MSE) between warped target landmark positions across intermediate timesteps. For each landmark pair, we sample 10 evenly spaced values of $t \in [0, 1]$, warp the landmarks accordingly, and compute their MSE, averaging the results across all timesteps. A

perfect alignment would yield an average MSE of $0$, since source and target landmarks would coincide exactly at every value of $t$. Experiments are conducted on three benchmarks: face / monument / 3D Gaussian avatar alignment.

**Sinusoidal vs. non-sinusoidal activations.** We compare NCF- and NODE-based variants of FLOWING using SIREN activations and their sigmoid counterparts to justify our architectural choices for high-frequency representations in morphing quality and convergence. Table 1 reports the average MSE across the FRLL, MegaDepth, and NeRSemble datasets for face warping, monument alignment, and 3D Gaussian avatar morphing. Results are shown for ifmorph, NCF, and NODE. For NCF and NODE, we evaluate both SIREN and sigmoid activations. Note that NCF (SIREN) and NODE (SIREN) outperform ifmorph by one to three orders of magnitude, while SIREN activations consistently yield lower errors than the sigmoid case. In particular, NODEs with sigmoid ("vanilla" configurations) perform significantly worse, confirming the importance of sinusoidal activations for accurate warping alignment.

Table 1: Results for the landmark alignment across the FRLL [10], MegaDepth [25] and NeRSemble [22] datasets. The best, second and third best results for each dataset/model combination are shown in green, yellow, and orange, respectively.

| Model (Activation) | FRLL ($\downarrow$) | MegaDepth ($\downarrow$) | NeRSemble ($\downarrow$) |
|---|---|---|---|
| ifmorph [44] | 1.5E-3 | 2.9E-1 | 1.0E-3 |
| NCF (sigmoid) | 8.7E-3 | 2.0E+1 | 7.2E-5 |
| NCF (SIREN) | 3.9E-5 | 4.2E-4 | 7.3E-5 |
| NODE (sigmoid) | 6.2E-2 | 1.9E-1 | 5.0E-4 |
| NODE (SIREN) | 1.4E-4 | 4.9E-4 | 5.9E-5 |

**NCF vs. NODE.** Figure 4 shows that NCFs and NODEs achieve better results with significantly fewer training steps compared to ifmorph. While NODEs converge quickly, they typically show poorer feature alignment. In contrast, NCFs require more steps to converge, resulting in better feature alignment overall. However, due to minor pixel-level variations between source and target features, this does not necessarily translate to noticeable visual differences.

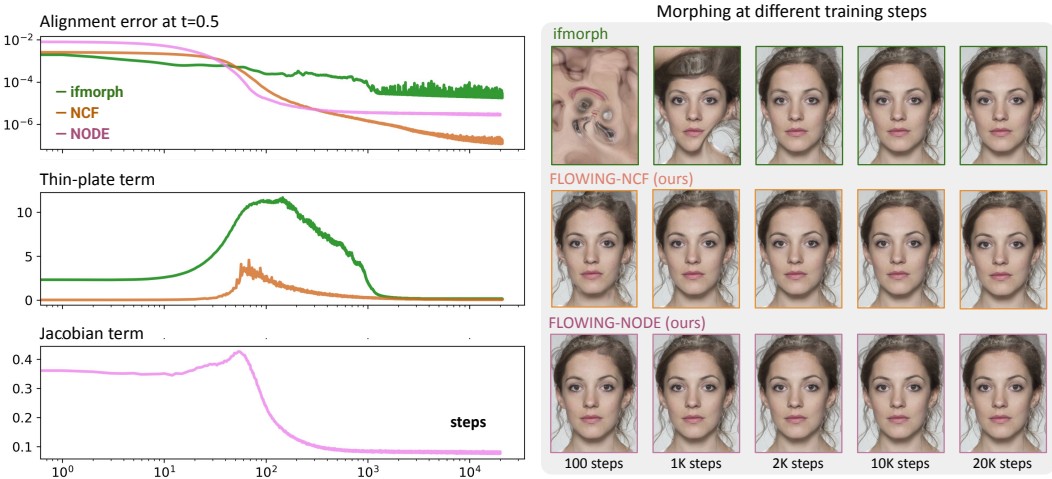

Figure 4: Convergence analysis of ifmorph (green), NCF (orange), and NODE (violet). Left: alignment and deformation metrics over training steps. For NODE, alignment is measured using the Jacobian term, while ifmorph and NCF use the Hessian-based thin-plate term. Right: qualitative morphing results at $t = 0.5$ after 100, 1000, 2000, 10000, and 20000 training steps. NCF and NODE achieve accurate alignment by 1000 steps, whereas ifmorph requires at least 2000 steps.

### 4.1.2 Blending quality

We evaluate blending quality using FRLL [10] for linear face morphing, following the same experimental setup as in the landmark alignment experiment (Sec. 4.1.1). We compare FLOWING with SIREN activations against ifmorph and a standard OpenCV baseline using Delaunay triangulation warping with linear blending[1]. Additional comparisons with the classical thin-plate spline method are given in Appendix F, and further experiments on video interpolation are discussed in Appendix E.

**Metrics.** Perceptual quality is measured using the *learned perceptual image patch similarity* (LPIPS) metric [60], computed between intermediate frames at $t = 0.5$ and both source and target images. We also measure the *Fréchet inception distance* (FID) [16] between generated morphs and original images, using 2048-dimensional feature vectors.

Table 2: Linear image blending results on the FRLL dataset [10]. The best, second-best, and third-best results are highlighted in green, yellow, and orange, respectively.

| Morphing type | LPIPS($I^0, I$) ($\downarrow$) | LPIPS($I, I^1$) ($\downarrow$) | FID ($\downarrow$) |
|---|---|---|---|
| OpenCV | 0.233 | 0.236 | 32.426 |
| ifmorph [44] | 0.250 | 0.252 | 38.427 |
| NCF (Ours) | 0.221 | 0.224 | 33.300 |
| NODE (Ours) | 0.210 | 0.213 | 31.952 |

Table 2 shows that both NCF and NODE outperform ifmorph across all metrics. NCF and NODE also obtained better results than OpenCV in LPIPS. For FID, NODE achieves the best overall performance, while NCF remains competitive, slightly below OpenCV. Overall, NODE produces higher-quality and more consistent morphing results than NCF.

### 4.1.3 Warping and morphing times

Table 3 summarizes training and morphing times obtained on an RTX 4090 GPU for both warp training and morphing (warping inference + blending). We report NODE with both 2000 and 20000 training steps to highlight its fast convergence. As shown, training times for NODE and ifmorph remain stable across different landmark counts, whereas NCF exhibits increased training time with additional landmarks. Morphing time depends on image resolution: NODE provides faster training but slower morphing, while NCF has slower training and intermediate morphing speed.

### 4.2 Image and face morphing

As shown in the previous sections, FLOWING can be used to warp between two images given a set of landmark correspondences. The warped images can then be blended using various techniques, such as linear blending, Poisson image editing [37], or generative blending [59]. In this work, we focus primarily on

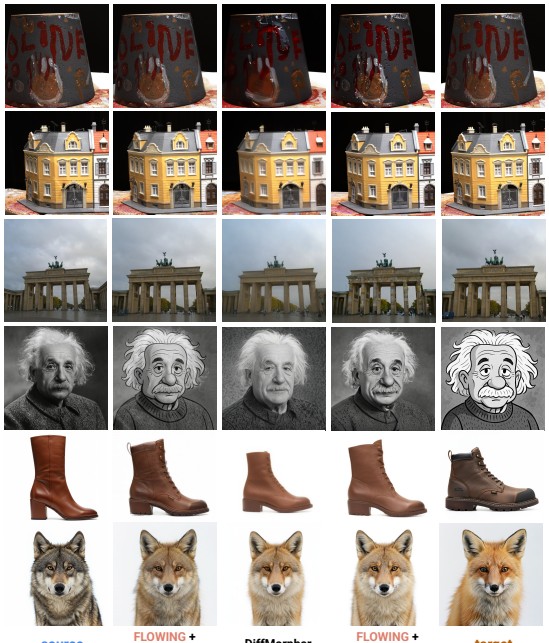

Figure 5: Applications of FLOWING to view interpolation (Rows 1-3), and stylization and object morphing (Rows 4-6). FLOWING produces smooth transitions even under environmental variations, maintaining geometric and photometric consistency. When combined with DiffMorpher, it further enhances structural coherence and visual fidelity, recovering details missed by DiffMorpher alone, such as the missing chimneys and the loss of column detail (Rows 2-3).

linear blending and generative blending via DiffMorpher [59]. Rows 1–3 of Figure 5 illustrate the view interpolation task, where FLOWING produces smooth transitions while better preserving both

---

[1] https://github.com/Azmarie/Face-Morphing/

geometric and photometric consistency across viewpoints. These examples show the benefit of applying FLOWING prior to generative blending in contrast to the usage of generative blending on its own. It improves both the consistency of the scene structure and reduces the number of missing elements or details. For instance, the building chimneys (Row 2) and architectural details (Row 3) remain intact, unlike in competing approaches. Rows 4–6 demonstrate stylization and object morphing, where combining FLOWING with generative models enhances structural coherence and visual fidelity across diverse visual domains, for example, in the photo-to-cartoon Einstein interpolation.

Table 3: Warp training times (68 and 130 landmarks) and morphing times at resolutions $256^2$ and $1350^2$. NODE is reported for both 2k and 20k training steps to highlight its fast convergence.

| Method | Steps | Warping training | | Morphing time | |
| | | 68 landmarks | 130 landmarks | Res. $256^2$ | Res. $1350^2$ |
| --- | --- | --- | --- | --- | --- |
| OpenCV | – | – | – | 0.01s | 0.06s |
| ifmorph [44] | 20k | 05m26s | 05m22s | 0.05s | 0.10s |
| NCF (Ours) | 20k | 08m48s | 15m05s | 0.02s | 0.31s |
| NODE (Ours) | 2k | 00m16s | 00m15s | 0.03s | 1.48s |
| NODE (Ours) | 20k | 02m39s | 02m38s | 0.03s | 1.48s |

Figure 6 shows the application of FLOWING to non-aligned, in-the-wild face images. Using the NODE backbone for warping and DiffMorpher for blending, our method produces morphings that are both visually coherent and perceptually realistic. Additional examples of generative morphing results are provided in Appendix D.

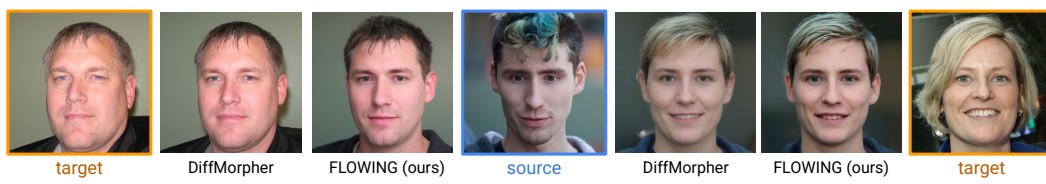

target    DiffMorpher    FLOWING (ours)    source    DiffMorpher    FLOWING (ours)    target

Figure 6: Qualitative results on in-the-wild faces from the FFHQ dataset [18]. By applying our flow-based warping prior to generative blending, FLOWING produces morphings that are significantly more coherent and realistic than those generated by DiffMorpher.

### 4.3 3D face morphing using Gaussian splatting

In this section, we evaluate our 3D face morphing framework using GaussianAvatars [39], an extension of 3DGS for photorealistic human head representation. Each face in GaussianAvatars is associated with a FLAME model [24], which enables the extraction of 3D facial landmarks spatially aligned with the Gaussian distribution. To morph between two GaussianAvatars, we apply the warping and blending formulation defined in (8). Figure 7 provides an overview of FLOWING applied to 3DGS morphing. On the left, we visualize the 3D flow field between two subjects derived from their landmarks and illustrate how the warped Gaussians can be combined to form an intermediate representation at $t = 0.5$, blending structural and appearance features from both faces. The middle panel shows additional morphing results for different subjects across multiple time steps. Our method achieves smooth and geometrically consistent transitions between subjects with distinct facial structures and appearance attributes. In particular, it naturally handles complex variations such as hair (second and third rows), thanks to the volumetric nature of 3DGS.

Finally, the right panel compares different blending strategies. The top row corresponds to a purely 2D setting, where warping is applied to the rendered images followed by linear blending, which results in severe misalignments and ghosting artifacts. In contrast, the middle and bottom rows show our 3D warping results, first with linear blending of the rendered images, and then with direct blending of the 3D Gaussians. The fully 3D pipeline produces smoother and more coherent transitions, effectively eliminating ghosting and preserving fine structural details across viewpoints. Additional qualitative examples of 3D morphing results are presented in Appendix C.

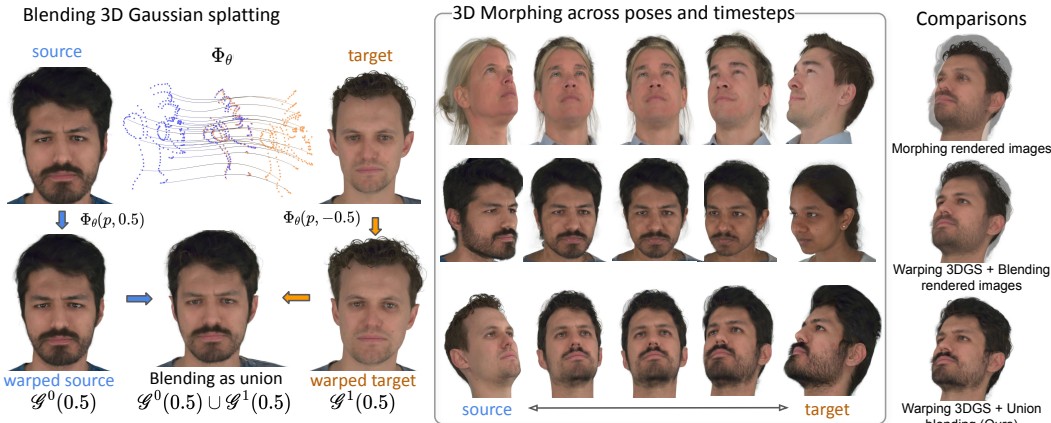

Figure 7: Overview of FLOWING for morphing between 3D faces using Gaussian splatting. Our approach warps the space to enforce 3D landmark alignment and applies union-based 3DGS blending, directly combining Gaussians in 3D space (left). This yields geometrically consistent morphs and preserves photorealistic appearance across poses and viewpoints (middle). The right panel compares blending strategies, showing that our union-based 3DGS blending produces smoother and more coherent results than morphing rendered images or blending after warping.

# 5    Conclusion and limitations

This work introduced FLOWING, a novel framework for morphing between graphical objects using flow-structured INRs. By leveraging the mathematical properties of flows, our approach enables faster convergence, more stable training, and robust warping behavior that avoids catastrophic singularities. We demonstrated the versatility of FLOWING across a wide range of tasks—including face morphing, view interpolation, and Gaussian-based 3D morphing—achieving high-quality and temporally coherent results. Moreover, FLOWING can be seamlessly integrated into generative pipelines as a replacement for traditional alignment procedures, producing morphings comparable to state-of-the-art generative approaches.

Despite these advantages, FLOWING inherits certain limitations intrinsic to flow-based formulations. Since it enforces invertibility by construction, it cannot model transformations involving occlusions or topological changes, such as morphing between faces with open and closed mouths. In such cases, generative models can complement our approach during the blending stage to recover missing structures. Additionally, as in other warping-based techniques, FLOWING depends on the quality and accuracy of landmark correspondences, whether extracted manually or through automated detectors.

As future work, we plan to extend FLOWING to point cloud alignment and registration, and further improve our Gaussian morphing approach.

# Acknowledgments

Guilherme and Nuno would like to thank Fundação de Ciência e Tecnologia (FCT) projects UIDB/00048/2020[2] and UIDP/00048/2020 for partially funding this work. Guilherme would also like to thank FCT project 2024.07681.IACDC[3] for partially funding this work. João Paulo would like to thank Fundação Carlos Chagas Filho de Amparo à Pesquisa do Estado do Rio de Janeiro (FAPERJ) grant SEI-260003/012808/2024 for funding this work. Vitor and Daniel gratefully acknowledge support from CAPES, grants 88887.842584/2023-00 and 88887.832821/2023-00, respectively for supporting this research. João Pereira is thankful for a start-up grant from the University of Georgia. We also thank Google for funding this research.

---

[2] DOI: `https://doi.org/10.54499/UIDB/00048/2020`

[3] DOI: `https://doi.org/10.54499/2024.07681.IACDC`

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

## A   Forward differentiation

We implement forward differentiation (FD) to speed up the calculation of Jacobian or Hessian terms, for thin-plate regularization. Forward mode differentiation is an alternative way to implement automatic differentiation, opposite to backward (or reverse mode) differentiation. FD is typically implemented using tangent (or dual) numbers [29]. Let us illustrate FD with an example. Suppose we want to evaluate $\frac{\partial x_n}{\partial x_0}$, where

$$x_i = f_i(x_{i-1}), \quad i = 1, \ldots, n. \tag{9}$$

Using the chain rule, we obtain that

$$\frac{\partial x_n}{\partial x_0} = \prod_{i=1}^{n} \frac{\partial x_i}{\partial x_{i-1}} = \prod_{i=1}^{n} f_i'(x_{i-1}).$$

To implement FD, we introduce the tangent (or dual) variables $\dot{x}_k$, $k = 0, \ldots, n$, defined by the recursion $\dot{x}_0 = \frac{\partial x_0}{\partial x_0} = 1$, and

$$\dot{x}_i = f_i'(x_{i-1})\dot{x}_{i-1}, \quad i = 1, \ldots, n \tag{10}$$

Using induction, it follows that

$$\dot{x}_k = f_k'(x_{k-1})\dot{x}_{k-1} = \prod_{i=1}^{k} f_i'(x_{i-1}) = \frac{\partial x_k}{\partial x_0}, \quad k = 1, \ldots, n.$$

Finally, the last element yields the desired derivative $\dot{x}_n = \frac{\partial x_n}{\partial x_0}$. To implement this approach, we can replace every function $f_i$ by an FD version, that calculates both $x_i$ and $\dot{x}_i$ at the same time.

Effectively, the FD version of $f_i$ maps ordered pairs $(x_{i-1}, \dot{x}_{i-1})$ to ordered pairs $(x_i, \dot{x}_i)$, so that both (9) and (10) are satisfied:

$$(x_i, \dot{x}_i) = f_i^{\text{FD}}(x_{i-1}, \dot{x}_{i-1}) := (f_i(x_{i-1}), f_i'(x_{i-1})\dot{x}_{i-1}), \quad i = 1, \ldots, n$$

Since we need to calculate Hessians, we develop a multivariate FD implementation involving ordered triplets $(z, \dot{z}, \ddot{z})$. Here, $\dot{z}$ contains intermediary gradients, and $\ddot{z}$ contains intermediary Hessians, with respect to a variable $x$. More specifically, if $z \in \mathbb{R}^m$ and $x \in \mathbb{R}^n$, our implementation ensures that $\dot{z} = \nabla_x z \in \mathbb{R}^{m \times n}$ and $\ddot{z} = \text{Hess}_x(z) \in \mathbb{R}^{m \times n \times n}$.

To calculate the Hessian of the model $f_\theta(x)$ with respect to $x$, we first write it as a composition of simpler functions (linear operators, non-linear activations, flow operator of affine flows), for which we have implemented forward differentiation: $f_\theta = f_1 \circ f_2 \circ \cdots \circ f_m$. Then, noting that $\nabla_x x = \mathbf{I}_n$ (the $n \times n$ identity matrix) and $\text{Hess}_x(x) = \mathbf{0}_{n \times n \times n}$, we calculate

$$f_\theta^{\text{FD}}(x, \mathbf{I}_n, \mathbf{0}_{n \times n \times n}) = f_1^{\text{FD}} \circ \cdots \circ f_m^{\text{FD}}(x, \mathbf{I}_n, \mathbf{0}_{n \times n \times n}) = (u, \dot{u}, \ddot{u}),$$

and set $\text{Hess}_x(f_\theta(x)) = \ddot{u}$.

Followingly, we provide the details for the FD implementation of linear transformations, non-linear activations, and the flow operator of an affine flow. By composing these, we can construct all architectures we consider in the paper.

**Linear operators:** Consider the linear operator $\mathcal{A}\mathbf{z} \mapsto \mathbf{A}\mathbf{z} + \mathbf{b}$. Since derivatives are also linear operators, the FD implementation is straightforward.

$$\mathcal{A}^{\text{FD}}(\mathbf{z}, \dot{\mathbf{z}}, \ddot{\mathbf{z}}) = (\mathbf{A}\mathbf{z} + \mathbf{b}, \mathbf{A}\dot{\mathbf{z}}, \mathbf{A} \times_1 \ddot{\mathbf{z}}),$$

where $\mathbf{A} \times_1$ denotes multiplication by $\mathbf{A}$ in the first dimension.

**Non-linear activations:** Suppose that $z \in \mathbb{R}$, $g : \mathbb{R} \to \mathbb{R}$, $\dot{z} \in \mathbb{R}^n$ and $\ddot{z} \in \mathbb{R}^{n \times n}$. Then using the chain rule, one obtains:

$$\begin{aligned}
g^{\text{FD}}(z, \dot{z}, \ddot{z}) &= (g(z), \nabla g(z), \text{Hess}(g(z))), \\
&= (g(z), g'(z)\dot{z}, \nabla(g'(z)\dot{z})), \\
&= (g(z), g'(z)\dot{z}, g'(z)\ddot{z} + g''(z)\dot{z}\dot{z}^T).
\end{aligned}$$

Here, the terms $g(z), g'(z)$ and $g''(z)$ are the derivatives of the activation function. For instance, for SIRENs, we have $g(z) = \frac{1}{w_0}\sin(w_0 z)$, $g'(z) = \cos(w_0 z)$ and $g''(z) = -w_0\sin(w_0 z) = -w_0^2 g(z)$.

**Flow operator of an Affine Flow:** The formula for an affine flow is provided in [6].

$$\Phi(\mathbf{x}, t) = e^{\mathbf{A}t}\mathbf{x} + \int_0^t e^{\mathbf{A}\tau}\mathbf{b}\, d\tau,$$

where the affine integral term is calculated exactly using an augmentation trick [6]. This flow has some nice properties: the map is linear on $\mathbf{x}$ and the derivatives of the exponential in terms of $t$ are easier to calculate. Suppose we want to evaluate $\Phi_{\text{FD}}(\langle \mathbf{x}, \dot{\mathbf{x}}, \ddot{\mathbf{x}} \rangle, \langle t, \dot{t}, \ddot{t} \rangle)$, where $\mathbf{x} \in \mathbb{R}^d$, $\dot{\mathbf{x}} \in \mathbb{R}^{d \times n}$, $\ddot{\mathbf{x}} \in \mathbb{R}^{d \times n \times n}$, $t \in \mathbb{R}$, $\dot{t} \in \mathbb{R}^n$ and $\ddot{t} \in \mathbb{R}^{n \times n}$. Letting $\mathbf{f} = \Phi(\mathbf{x}, t)$, $\dot{\mathbf{f}} = \mathbf{A}\mathbf{f} + \mathbf{b}$ and $\mathbf{E} = e^{\mathbf{A}t}$, we obtain, using the chain rule:

$$\begin{aligned}
\Phi^{\text{FD}}((\mathbf{x}, \dot{\mathbf{x}}, \ddot{\mathbf{x}}), (t, \dot{t}, \ddot{t})) &= (\Phi(\mathbf{x}, t), \nabla_x\Phi\dot{\mathbf{x}} + \Phi_t \dot{t}^T, \nabla_x^2\Phi\ddot{\mathbf{x}} + \nabla_x\Phi_t \otimes \dot{t} + \Phi_{tt} \otimes \dot{t}), \\
&= (\mathbf{f}, \mathbf{E}\dot{\mathbf{x}} + \dot{\mathbf{f}}\dot{t}^T, \mathbf{E} \times_1 \ddot{\mathbf{x}} + (\mathbf{AE}\dot{\mathbf{x}}) \otimes \dot{t} + (\mathbf{A}\dot{\mathbf{f}}) \otimes \dot{t}).
\end{aligned}$$

# B  Ablation studies

**NODE integration steps.** Neural ODEs require solving differential equations through numerical integration, both during training and inference, to associate each point with its corresponding source and target coordinates by integrating forward and backward in time. Thus, it might appear that the trajectories need to be approximated using a large number of steps, making the process computationally expensive. However, we find that this is not the case: high-quality solutions can

be obtained with only a few steps. We sampled several initial conditions from the $[-1, 1]^2$ grid and displaced the points using the trained NODE model with the same number of integration steps used during training. We then approximated the reference (ground-truth) trajectories by computing a high-resolution baseline with 1000 integration steps. As a quality metric, we measured the maximum squared error between the predicted trajectory points and the linearly interpolated baseline points; a lower error indicates that the numerical integration used during training accurately captures the underlying dynamics. Table 4 shows the results of this experiment. We observe that only a small number of integration steps is sufficient to approximate the flow with high fidelity. Moreover, applying Jacobian regularization during training further stabilizes the dynamics, enabling reliable results even with very few integration steps.

Table 4: Comparison of trajectory quality (maximum squared error) of NODEs with different integration steps against a baseline of 1000 steps. NODE-based morphing needs very few steps in order to obtain high quality approximations of the associated flow.

| Int. steps | MegaDepth Sample | | FRLL Sample | |
| | No regularization | With regularization | No regularization | With regularization |
|---|---|---|---|---|
| 3 | 5.44E-05 | 6.43E-07 | 1.36E-07 | 7.16E-10 |
| 5 | 3.39E-08 | 3.27E-09 | 2.08E-10 | 6.19E-11 |
| 7 | 1.88E-09 | 5.24E-10 | 5.46E-11 | 6.00E-11 |
| 15 | 8.88E-12 | 5.18E-10 | 5.46E-11 | 6.10E-11 |

**Forward differentiation (FD).** We conducted an experiment to evaluate the computational speed-up achieved by our FD scheme for Hessian computation. An NCF model was initialized using the same configuration as in our main experiments. For this model, we computed the Hessian at 1,000 domain points using both PyTorch's `autograd` and our FD implementation, then evaluated the thin-plate loss and performed backpropagation. Each method was run 100 times, and we report the average computation time per iteration. On an NVIDIA GeForce RTX 4090 GPU, our FD implementation achieved an average iteration time of $6.93 \times 10^{-3}$ seconds, compared to $1.62 \times 10^{-1}$ seconds using `autograd`, corresponding to a $23.4\times$ speed-up. This demonstrates the substantial efficiency gains of our FD approach over standard automatic differentiation.

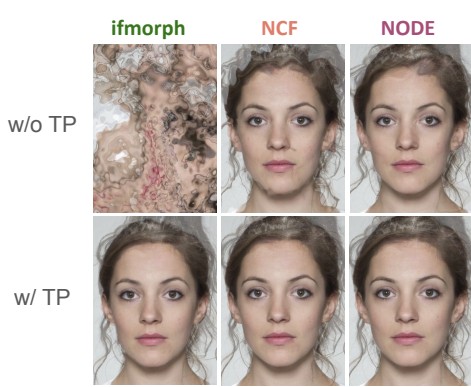

Figure 8: Effect of TP regularization on if-morph (left) and our methods, NCF (center) and NODE (right). The top row shows results without TP regularization, while the bottom row shows the same models with TP applied. TP regularization promotes smoother deformations and enhances structural coherence across the morphing process.

**Thin-plate regularization.** We perform an ablation study to evaluate the effect of thin-plate (TP) regularization on deformation smoothness and structural consistency. As illustrated in Figure 8, TP regularization significantly improves the coherence of the deformation field, reducing distortions and enforcing smoother transitions between source and target landmarks. The effect is most pronounced for the baseline ifmorph, which exhibits severe artifacts in the absence of regularization. In contrast, our models (NCF and NODE) already demonstrate stable deformation behavior due to their flow-based formulation, yet still benefit from TP regularization, yielding even more consistent and visually coherent morphings.

## C  Additional 3DGS morphing experiments

**Flow evaluation**  Figure 9 illustrates the 3D flow streamlines corresponding to landmark trajectories from the source image for three methods used in 3DGS morphing: ifmorph, and FLOWING with NCF and NODE backbones. As shown, ifmorph produces flow fields that are less stable and exhibit pronounced curvature, while NCF and NODE yield smoother, more coherent, and regularized flows.

This difference is particularly evident in regions not directly constrained by landmarks—such as the hair—where ifmorph introduces noticeable distortions. In contrast, the FLOWING variants preserve both structural continuity and spatial consistency across the field.

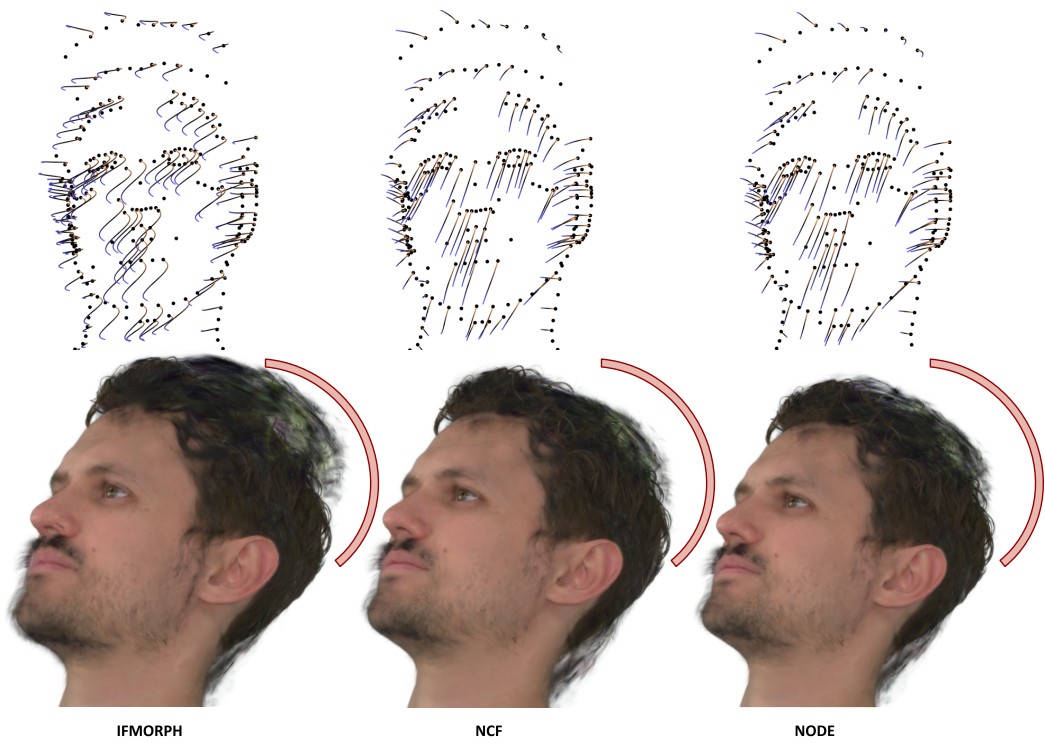

Figure 9: Visualization of 3D flow fields generated by ifmorph, NCF, and NODE in the 3DGS morphing task. The red arch highlights instability in the hair region for ifmorph, whereas NCF and NODE preserve structure more reliably.

**Training details**    Our 3DGS morphing configuration extends the 2D morphing pipeline while maintaining the same loss weights. To accommodate the higher dimensionality, we use slightly larger learning rates (LRs) and an LR scheduler for both NODE and NCF to ensure convergence within 20,000 steps. The initial LRs are set to 0.001 for NODE, 0.002 for NCF, and 0.0001 for ifmorph. Early stopping is employed with a patience of 500 epochs for NCF and NODE, and 1,000 for ifmorph. If the loss plateaus for 100 epochs, the LR is reduced, and training terminates once the patience threshold is reached. The best-performing model is selected based on the lowest loss.

**Additional results**    Figure 10 presents additional qualitative results for our Gaussian morphing framework. FLOWING consistently achieves smooth and coherent transitions across diverse subjects, effectively handling challenging regions such as facial and head hair (third row). Furthermore, it demonstrates strong generalization when morphing between subjects of different genders, maintaining both geometric structure and photometric consistency.

## D    Additional comparisons using generative blending

Figure 11 shows additional qualitative results combining FLOWING with DiffMorpher as a generative blending strategy. These examples further confirm that introducing a warping stage prior to generative blending significantly improves the quality of the final morphs. In particular, FLOWING enhances spatial alignment and structural coherence, producing smoother intermediate representations, even in simple scenarios such as morphing between spherical objects.

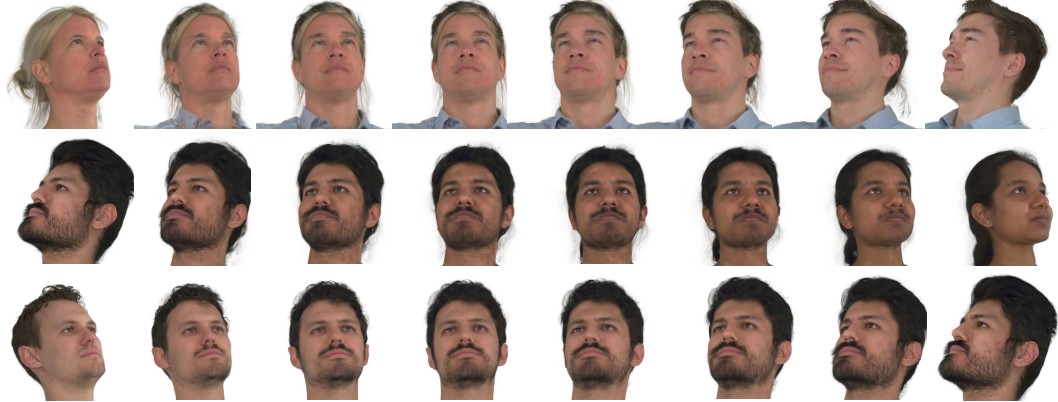

Figure 10: Additional results of Gaussian morphing across multiple timesteps (first and last columns show the targets). FLOWING achieves smooth and coherent transitions across diverse subjects while preserving structure and appearance.

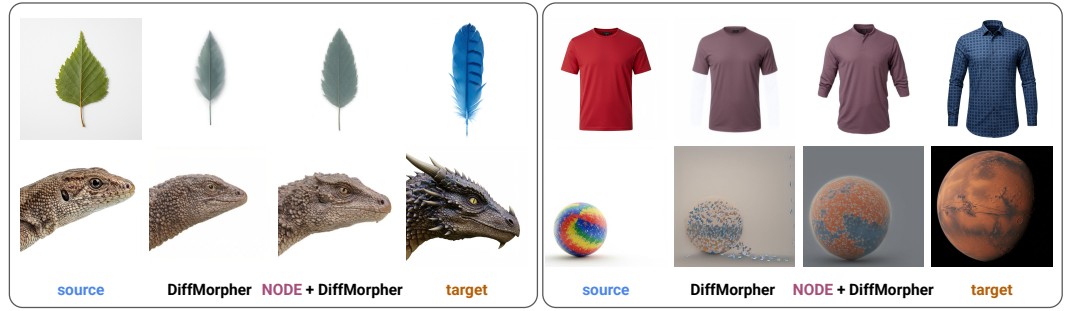

Figure 11: Additional examples comparing generative blending with and without FLOWING as a warping prior. Incorporating FLOWING before applying DiffMorpher (NODE+DiffMorpher, ours) yields superior structural preservation and smoother transitions.

## E   Applications in video interpolation

Video interpolation aims to synthesize intermediate frames between two given frames, a key component in applications such as video compression and frame rate upscaling. Most deep learning-based approaches rely on estimating optical flow between consecutive frames, which describes pixel motion as a vector field [17; 23]. Assuming temporal proximity between frames, optical flow can be leveraged to approximate intermediate frames by integrating a time-independent ODE defined by the flow field.

Inspired by this principle, we explore the use of FLOWING to learn a compact representation of optical flow from sparse samples and to reconstruct intermediate frames. For this experiment, we employ a cropped sequence from the Sintel dataset [8] and use RAFT [53] to generate dense optical flow between frames. We then sample a sparse set of image points and construct landmark pairs $(p^0, p^1)$ by displacing them according to the optical flow. FLOWING is trained to model a continuous flow field aligning these points over time. To evaluate interpolation quality, we warp and linearly blend the input frames at $t = 0.5$ using the learned flow and compare the resulting image against the ground-truth middle frame. We further compare against three baselines: (1) linear frame blending, (2) dense optical-flow warping followed by linear blending, and (3) ifmorph [44]. As shown in Table 5, FLOWING achieves lower MSE than all baseline interpolation methods, indicating improved accuracy in predicting intermediate frames. These preliminary results suggest that learning a continuous optical flow representation from sparse correspondences is a promising direction for video interpolation.

Table 5: Comparison of interpolation accuracy against the ground-truth intermediate frame on Sintel [8]. Lower MSE indicates better reconstruction.

| Method | MSE ($\downarrow$) |
|---|---|
| Linear Blend | 2.6E-4 |
| Optical Flow Warp | 4.9E-5 |
| ifmorph | 5.8E-5 |
| NCF (Ours) | **4.0E-5** |
| NODE (Ours) | **3.9E-5** |

## F  Comparison with TPS warping

Another standard morphing algorithm is thin-plate-spline warping. While TPS warping does not involve an iterative process, the method does require solving a linear system of equations in order to find the appropriate linear combination of the basis functions. The matrix of such a system can, in some cases, be badly conditioned. When using the scikit-image implementation of the algorithm, we find that this problem leads to crashes or low-quality warpings in approximately 10% of FRLL samples. This can negatively impact metrics such as the LPIPS. FLOWING, on the other hand, does not present these problems as it does not rely on solving a similar badly-conditioned system. For comparison, using TPS warping combined with linear blending for the FRLL dataset, we obtain $\text{LPIPS}(I^0, I) = 0.371$ and $\text{LPIPS}(I^1, I) = 0.374$.

