# OpenReview forum: "FLOWING: Implicit Neural Flows for Structure-Preserving Morphing"
_NeurIPS.cc/2025/Conference — NeurIPS 2025 poster_

### Official Review · Reviewer_h25A · 2025-06-30

**Clarity:** 3
**Significance:** 3
**Originality:** 3
**Rating:** 5
**Confidence:** 4

**Summary:**

This paper introduces a flow-based implicit neural representation to estimate the vector flow to morph a source into a target one. By casting the problem as learning a time-dependent diffeomorphism, this method avoids expensive regularization overheads and naturally guarantees continuity, invertibility and temporal coherence. This method can generalize to both 2D and 3D inputs, exhibiting its huge generalization ability of the framework. The training target consists of two simple terms, a data penalty to ensure temporal coherence and invertibility, and a regularization term on flow Jacobian to ensure continuity and smoothness. At inference, the learned flow warps images or 3D Gaussian‐splat representations, followed by linear or generative blending for smooth interpolation. The authors adapt Neural ODE and Neural Conjugate Flows to solve their defined problem, and both achieves reasonable results.

**Questions:**

- How much does this method rely on the accuracy of the correspondence pair?
- Due to the implicit formulation, I wonder how efficient this method is compared to other methods?
- In the 3D case, how many Gaussian blobs are used? What if the source Gaussians have a significantly different number of blobs from the target one?

**Ethical Concerns:**

["NO or VERY MINOR ethics concerns only"]

**Final Justification:**

The authors have addressed my questions and concerns. While the inference time is higher than other methods, I believe it can be optimized. Methodologically, framing morphing into INRs provides a new insight in this field. Moreover, the versatility of this method may push the method even further. I raise my score to accept.

**Limitations:**

Yes

**Quality:**

3

**Strengths And Weaknesses:**

### Strengths
- The method applies flow-based implicit neural representation to estimate the flow for warping. Since the formulation satisfies an ODE, the method naturally overcomes the difficulty of temporal coherence and continuity.
- Great versatility. This method can be solved by different architectures, and be applied to both 2D and 3D data, as well as both feature space and data (images or 3D points) space.
### Weakness
- No discussion on the computational efficiency.
- The definition of $\mathcal{F}^0$ at L162 seems wrong. For $i = 0$, the parameter should be $t$.
- Likewise to Equation. 8

---

> ### Author Rebuttal · Authors · 2025-07-30
>
> We thank the reviewer for the detailed comments. We are encouraged by the positive remarks regarding the fact that it naturally overcomes the difficulty of temporal coherence/continuity, and also regarding its great versatility with respect to architectures, 2D/3D data and feature/data space. We will try our very best to address the reviewer’s concerns by performing additional experiments regarding computational efficiency and Gaussian morphing quality with respect to the number of blobs.
> We are happy to provide additional clarifications as needed, during the discussion period.
>
> ## Weaknesses
> > **No discussion on the computational efficiency.**
>
> We did some additional experiments regarding memory/compute requirements, which we describe next.
> Memory requirements for FLOWING and ifmorph training are not heavily affected by landmark count, being around 2.5GB for FLOWING and around 1.5GB for ifmorph. Regarding compute requirements, we present next some experiments on a RTX 4090 GPU for both warp training and blending. We show results for NODE trained using both 2k and 20k training steps, since it converges quickly. We can see that NODE and ifmorph training times do not change significantly with respect to landmark count, which is not true for NCF. The blending times will depend on image resolution. NODE presents faster training and slower blending, while NCF has slower training and intermediate blending time.
>
> | Method | Landmark count | Training steps | Warp training time↓ |
> | ---------- | --------------------- | ------------------- | -------------------------- |
> | ifmorph | 68                      | 20k                  | 05m26s                    |
> | NCF (Ours) | 68                      | 20k                  | 08m48s                    |
> | NODE (Ours) | 68                      | 2k                    | 00m16s                    |
> | NODE (Ours) | 68                      | 20k                  | 02m39s                   |
>
> | Method | Landmark count | Training steps | Warp training time↓ |
> | ---------- | --------------------- | ------------------- | -------------------------- |
> | ifmorph | 130                      | 20k                  | 05m22s                    |
> | NCF (Ours) | 130                      | 20k                  | 15m05s                    |
> | NODE (Ours) | 130                      | 2k                    | 00m15s                    |
> | NODE (Ours) | 130                      | 20k                  | 02m38s                   |
>
> | Method | Image resolution | Morphing time (s)↓ |
> | ---------- | --------------------- | -------------------------- |
> |OpenCV | 256 x 256          | 0.01                   |
> | ifmorph | 256 x 256           | 0.05                    |
> | NCF (Ours) | 256 x 256    | 0.02                    |
> | NODE (Ours) | 256 x 256 | 0.03                   |
>
> | Method | Image resolution | Morphing time (s)↓ |
> | ---------- | --------------------- | -------------------------- |
> |OpenCV | 1350 x 1350          | 0.06                   |
> | ifmorph | 1350 x 1350           | 0.10                    |
> | NCF (Ours) | 1350 x 1350    | 0.31                    |
> | NODE (Ours) | 1350 x 1350 | 1.48                  |
>
> > **The definition of F^0 at L162 seems wrong. For i=0, the parameter should be t. Likewise to Equation 8**
>
> The relationship between the features, which flow forward in time for the source and backwards for the target, and the resulting pixel values of the warped image, which flow in the inverse direction, is somewhat unintuitive and will be better covered for the paper's final version.
>
> In essence, to construct a warped image, we reverse the direction of the flow, so as to identify the source of the pixel at hand. Take, for example, $i = 0$ and $t=0.5$. The pixel value at a given coordinate $x$ will be the pixel value of the point $x’$ in the original image which landed on $x$ after flowing for $0.5$ units of time. This can be written as follows:
> $$ \mathscr{I}^0(x,t) = I^0(x’), \text{ where } x = \Phi(x’,t) $$
>
> Now, because our flows are autonomous, the latter relationship may be inverted to express x’ in terms of x:
> $$x’ = \Phi(x,-t)$$
>
> Combining these expressions together, we land at the formula at L162:
>
> $$ \mathscr{I}^0(x,t) = I^0(\Phi(x, -t)) $$
>
> An analogous argument may be derived for the target image ($i = 1$).
>
> With the final version’s additional content page, we expect to be able to describe this relationship more clearly and intuitively.
>
> ## Questions
>
> > **How much does this method rely on the accuracy of the correspondence pair?**
>
> The method suffers from a significant, albeit not critical, reliance on the accuracy of corresponding pairs. In preliminary experiments, we have verified that small amounts of noise distributed along all features or large amounts of noise applied to few features can be mostly handled by the network, in part due to the regularizations implicit to the framework. In particular, uniqueness should “force” noisy features into their place when surrounded by accurate ones. The same should apply in the presence of small amounts of mislabeled pairs. This is, in some sense, the best possible outcome for a fundamentally feature-based method.
>
> > **Due to the implicit formulation, I wonder how efficient this method is compared to other methods?**
>
> The answer to this question depends on what type of efficiency is being discussed. One of the main features of INRs is the compact representations they provide for geometrical objects, meaning it is exceptionally memory efficient. Likewise, NODE is much more computationally efficient than other implicit representations, such as ifmorph.
>
> > **In the 3D case, how many Gaussian blobs are used? What if the source Gaussians have a significantly different number of blobs from the target one?**
>
> Models from Gaussian avatars typically have between 60,000 and 120,000 Gaussians, which represents a significant change in magnitude. Since alignment relies on 3D landmarks, it is independent of the number of Gaussians in the source and target. For example, in the supplementary video, the source has 108,142 Gaussians and the target 89,021—a considerable ~20k difference. We also conducted an experiment with the target downsampled to 73,939 Gaussians, increasing the difference to around 35k. The results were similar, though we can’t show them here due to image submission limits. They will be included in the paper’s Supplementary Material.

---

> > ### Author Response · Authors · 2025-08-06
> >
> > Dear Reviewer h25A,
> >
> > We appreciate the thorough review and constructive comments. We have submitted point-to-point responses to your questions and observations, including clarifications on the paper's notation as well as an additional experiment. We would appreciate it if you could inform us whether your concerns have been addressed. We are also happy to provide further clarification if needed.
> >
> > Best regards,
> >
> > Authors of paper #13820

---

> > ### Comment · Reviewer_h25A · 2025-08-06
> >
> > I thank the authors for their comments. Most of my concerns are addressed. While the inference time is higher than other methods, I believe it can be further optimized. The versatility of the proposed work is favored.

---

### Official Review · Reviewer_12rU · 2025-07-03

**Clarity:** 3
**Significance:** 1
**Originality:** 1
**Rating:** 3
**Confidence:** 3

**Summary:**

Given an annotated set of corresponding points in a fixed and moving image, this paper trains a neural network to interpolate between these corresponding points.

**Questions:**

The main comparison that I'm interested in is the comparison to the OpenCV baseline in table 2. To convince me to increase my score, the main avenue would be to convince me that OpenCV's warping algorithm is seriously well tuned for VPIP, and so this method is impressive for edging it out- for example, if you show that you did some hyperparameter search for the OpenCV baseline on this task, or that this is a benchmark task that the OpenCV authors were aware of and were optimizing for.

**Ethical Concerns:**

["NO or VERY MINOR ethics concerns only"]

**Limitations:**

yes

**Paper Formatting Concerns:**

I did not have formatting concerns- the formatting was well done.

**Quality:**

2

**Strengths And Weaknesses:**

It's interesting to visually see the flow field from when the pre-existing NODE and NCF algorithms are applied to 2-D points, although NCF in particular is being used as something of a chainsaw cutting tissue paper- the work in NCF was needed for interpolating in extremely high dimensional spaces such as pixel intensities of RGB images.

The practicality of this approach seems dubious to me. It seems like an enormous amount of machinery and computation to replace a thin plate spline or delauney triangulation: the improvement over the opencv baseline in table 2 seems marginal.

It would be more convincing if the authors had picked one of NCF or NODE- the two proposed approaches seem to have little in common to make this a single paper.

The improvement over the INR baseline is dramatic and real- if you already have a correspondence oracle but are absolutely determined that the correspondence between the remaining points in the image will be achieved by training an MLP, then this approach is correct.

---

> ### Author Rebuttal · Authors · 2025-07-30
>
> We thank the reviewer for the detailed comments. We are encouraged by positive remarks regarding dramatic and real improvement over the INR baseline. We performed additional experiments regarding comparisons with the OpenCV baseline in challenging scenarios.
>
> ## Strengths and Weaknesses
>
> > **[...]NCF in particular is being used as something of a chainsaw cutting tissue paper- the work in NCF was needed for interpolating in extremely high dimensional spaces such as pixel intensities of RGB images.**
>
> We believe the reviewer refers here to Continuous Normalizing Flows (CNF) [2,3], which do take as inputs high-dimensional pixel data. However, the method we use in the paper are Neural Conjugate Flows (NCF) [1], which had first been used in the context of low-dimensional dynamical systems, for example 2D/3D systems.
>
> Though the names (and especially acronyms) of each architecture are similar, their scope is quite different. NCFs, at least as originally proposed, usually take as input coordinates in state or image space, like we use in the paper.
>
>
> [1] Bizzi, Arthur, Lucas Nissenbaum, and João M. Pereira. "Neural Conjugate Flows: A Physics-Informed Architecture with Flow Structure." Proceedings of the AAAI Conference on Artificial Intelligence. Vol. 39. No. 15. 2025.
>
> [2] Chen, Ricky TQ, et al. "Neural ordinary differential equations." Advances in neural information processing systems 31 (2018).
>
> [3] Grathwohl, Will, et al. "FFJORD: Free-Form Continuous Dynamics for Scalable Reversible Generative Models." International Conference on Learning Representations. 2019.
>
> > **[...] It would be more convincing if the authors had picked one of NCF or NODE- the two proposed approaches seem to have little in common to make this a single paper.**
>
> As mentioned above, both NCFs and Neural ODEs operate quite similarly - NCFs are originally introduced as “integration-free” alternatives to Neural ODEs. Our intention by including them was to demonstrate that the framework is, in fact, flexible and does not necessarily depend on the specific architecture, as long as it displays the structure of flows.
>
> > **The practicality of this approach seems dubious to me. It seems like an enormous amount of machinery and computation to replace a thin plate spline or delauney triangulation: the improvement over the opencv baseline in table 2 seems marginal.**
>
> Unlike OpenCV, our method avoids strictly straight paths, resulting in better feature alignment. Importantly, it can enforce the uniqueness of trajectories, avoiding ghosting; an artifact common when straight lines intersect during interpolation. While we cannot include a figure illustrating this, we refer to Figure 9 of the ifmorph paper, which depicts a similar issue.
>
> Likewise, classical methods like thin-plate splines and Delaunay triangulation work in 2D but do not generalize well to higher dimensions, nor do they integrate with neural pipelines. In contrast, our flow-based model is fully differentiable, dimension-agnostic (2D/3D), and integrates seamlessly with neural modules like INRs and generative tools.
>
> ## Questions
>
> > **The main comparison that I'm interested in is the comparison to the OpenCV baseline in table 2. To convince me to increase my score, the main avenue would be to convince me that OpenCV's warping algorithm is seriously well tuned for VPIP, and so this method is impressive for edging it out- for example, if you show that you did some hyperparameter search for the OpenCV baseline on this task, or that this is a benchmark task that the OpenCV authors were aware of and were optimizing for.**
>
> OpenCV’s warping method is based on Delaunay triangulation and, to the best of our knowledge, does not have tunable hyperparameters for morphing. However, due to its maturity and widespread use, it serves as a strong classical baseline. To evaluate performance in more challenging scenarios, we conducted an experiment using unaligned faces from Figs. 7 and 9 of the ifmorph paper. In this case, our proposed NCF and NODE models both outperform OpenCV and ifmorph, with NODE achieving the best results:
>
> | Morphing Type | LPIPS(I₀, I)↓ | LPIPS(I, I₁)↓ |
> | -------------------- | ---------------- | ---------------- |
> | OpenCV           | 0.408           | 0.421            |
> | ifmorph             | 0.421           | 0.408           |
> | NCF (Ours)      | 0.391           | 0.392           |
> | NODE (Ours)   | **0.375**      | **0.377**      |
>
>  We are happy to provide additional clarifications as needed, during the discussion period.

---

> > ### Author Response · Authors · 2025-08-06
> >
> > Dear Reviewer 12rU,
> >
> > We appreciate the thorough review and constructive comments. We have submitted point-to-point responses to your questions and observations, including clarifications on the role of NCFs and how FLOWING compares to OpenCV, as well as an additional experiment. We would appreciate it if you could inform us whether your concerns have been addressed. We are also happy to provide further clarification if needed.
> >
> > Best regards,
> >
> > Authors of paper #13820

---

### Official Review · Reviewer_Bay7 · 2025-07-04

**Clarity:** 3
**Significance:** 3
**Originality:** 2
**Rating:** 3
**Confidence:** 3

**Summary:**

This paper proposes FLOWING, a framework for morphing between 2D images and 3D Gaussian splats using flow-based implicit neural representations. The method leverages Neural ODEs and Neural Conjugate Flows to ensure continuity, invertibility, and temporal coherence by construction, avoiding expensive regularizations used in prior work like ifmorph. The approach demonstrates applications in face morphing, image interpolation, and 3D Gaussian Splatting morphing.

**Questions:**

- How does FLOWING perform on datasets lacking clearly defined landmark structures, such as natural scenes with heavy occlusion or clutter, or on domains outside conventional faces (e.g., animals, medical images)? Is it possible to relax landmark reliance?
- Could you quantify and compare compute/memory requirements and wall-clock times for the main approaches (FLOWING vs. ifmorph vs. classical), especially as input/image sizes or landmark counts scale?
- How does FLOWING compare to classical, non-learning-based flow warping (e.g., dense optical flow, thin-plate spline warping), both quantitatively and qualitatively on the main datasets?

**Ethical Concerns:**

["NO or VERY MINOR ethics concerns only"]

**Final Justification:**

In authors' final remark, they said:
"Novelty of Sinusoidal INRs + Neural ODEs – Acknowledged the employment of these techniques in Sun et al. (2024), in the context of medical registration; highlighted that our work is the first to adapt NODEs and NCFs with sinusoidal INRs to morphing of images and 3DGS, addressing morphing-specific constraints and demonstrating strong results."

I didn't see substantial technical contributions from adapting the same paradigm into images and 3DGS given Sun et al. (2024) has already dealt with 3D image registration problem.

I acknowledge their efforts in comprehensive experiments. My final rating is made based on the limited technical novelty.

**Limitations:**

Yes; limitations are discussed—primarily in acknowledgment of ablation experiments and practical constraints—but further explicit analysis of error/failure cases and dataset diversity would strengthen the discussion.

**Paper Formatting Concerns:**

The paper is technically sound but the evaluation is not sufficiently broad or deep to justify acceptance; reasons to reject slightly outweigh reasons to accept.

**Quality:**

3

**Strengths And Weaknesses:**

**Strengths**:
- Quantitative results in Table 1 and Table 2 show striking improvements: for example, mean squared alignment errors drop by orders of magnitude on the FRLL and MegaDepth datasets when using the SIREN-activated NCF/NODE models compared to ifmorph. LPIPS and FID scores for image blending in Table 2 further highlight the superiority of FLOWING for perceptual quality.
- FLOWING converges much faster and with more stability than ifmorph, which is susceptible to singularities and unstable warps.
- Versatility for 2D and 3D
- The method can be augmented by generative blending (e.g., DiffMorpher) after flow-based alignment, yielding visually superior morphs in diverse scenarios.

**Weaknesses**
- Flow-based morphing is widely applied in other domains, such as medical image registration (Voxelmporph-diff). Neural ODE-based solutions are also well-explored, such as NODEO (CVPR22), Neural Diffeomorphic Flow (CVPR22) and NMF (NeurIPS 2020). It is unclear what novel theoretical contributions are beyond prior works.
- Though the landmark-based formulation is effective for faces or some designed datasets, the approach's reliance on accurate correspondences may limit applicability to scenarios where dense or reliable landmark annotations are unavailable or ambiguous, such as unconstrained scenes or many non-human objects.
- The claim that minimizing Jacobian/Hessian norms produces plausible, semantically reasonable morphs is plausible but not universally justified, especially in settings where "wiggly" but valid transitions may be preferred in some tasks.
- No Comparison with Learning-Free Flow Methods: Method comparisons seem limited to ifmorph and OpenCV warping. The inclusion of established, learning-free or classical flow-based approaches (e.g., dense optical flow with smoothing) would help contextualize FLOWING’s place among "classic methods" and modern INR-based approaches.

---

> ### Author Rebuttal · Authors · 2025-07-30
>
> We would like to thank the reviewer for the detailed and constructive comments on our work. We are encouraged by the remarks regarding the quantitative metrics, as well as by the mentions of perceptual quality, fast convergence and versatility. We hope our response can address the reviewer’s concerns. We conducted additional experiments regarding comparisons with classical/non-learning-based flow warping, and also regarding compute/memory requirements. If there are any remaining questions, we are more than happy to address them.
>
> ## Weaknesses
>
> > **It is unclear what novel theoretical contributions are beyond prior works (Voxelmporph-diff, NODEO, Neural Diffeomorphic Flow, NMF).**
>
> We appreciate these references and intend to add them to the final paper. Although these architectures can be said to share our objectives of using flow-like transformations that preserve geometrical properties, they do so in different contexts with different tools.
>
> The main theoretical tools we develop are tailored specifically to the interplay of implicit neural representations (INRs) and morphing. In particular, we demonstrate how the group properties of flows translate to desirable properties of morphing transformations. To the best of our knowledge, this connection has not been made explicit before.
>
> In contrast, Voxelmorph and NODEO generally deal with the carrying of signals in high-dimensional spaces with CNNs, in a setup more akin to optimal transport. Among other things, this makes them inadequate for temporal feature matching, as well as blending between signals with distinct color content. Likewise, the approaches of NDF and NMF are built on different latent representations and therefore cannot be easily used in our applications.
>
> Our contributions are also methodological: To the best of our knowledge, combining sinusoidal INRs and Neural ODEs has not been explored in prior work.
>
> > **Though the landmark-based formulation is effective for faces or some designed datasets, the approach's reliance on accurate correspondences may limit applicability to scenarios where dense or reliable landmark annotations are unavailable or ambiguous, such as unconstrained scenes or many non-human objects.**
>
> We acknowledge this limitation and discuss it in the supp mat. However, even generative methods often implicitly rely on alignment of features—for example, DiffAE requires pre-aligned inputs, and DiffMorpher struggles with misaligned cases (see Fig. 4 in the supplementary material). Easing these restrictions is a major avenue for future work.
>
> > **The claim that minimizing Jacobian/Hessian norms produces plausible, semantically reasonable morphs is plausible but not universally justified, especially in settings where "wiggly" but valid transitions may be preferred in some tasks.**
>
> Our TPS regularization reduces distortion and enforces smoothness, but it is optional. As shown in Fig. 1 of the supplementary material, disabling the TPS term in FLOWING still produces plausible morphs with wiggly deformations while preserving keypoint alignment. This flexibility is not present in prior INR methods like ifmorph, where TPS is essential due to the absence of true flow structure. Since we are not able to use figures in the rebuttal, we will include in the final version an experiment illustrating how varying the TPS weight affects the deformation behavior.
>
> The reviewer’s comment suggests a promising avenue for future work: designing regularizations that impose physical or task-specific constraints on the vector field, such as deformations with inertia (“jelly-like”).
>
> > **No Comparison with Learning-Free Flow Methods: Method comparisons seem limited to ifmorph and OpenCV warping. The inclusion of established, learning-free or classical flow-based approaches (e.g., dense optical flow with smoothing) would help contextualize FLOWING’s place among "classic methods" and modern INR-based approaches.**
>
> Thank you for the suggestion. Classical optical flow methods typically assume small motion between consecutive frames of the same scene, which differs from our morphing setting involving large geometric and semantic changes. Nonetheless, we agree that classical baselines help contextualize FLOWING, and we include a comparison with thin-plate spline (TPS) warping on a subset of the FRLL dataset. TPS is applied using landmark interpolation followed by linear blending, and we report LPIPS metrics below for comparison:
>
> | Morphing Type | LPIPS(I₀, I)↓ | LPIPS(I, I₁)↓ |
> | -------------------- | ---------------- | ---------------- |
> | TPS warping    | 0.311           | 0.314            |
> | NODE (Ours)   | **0.210**      | **0.213**      |
>
> These results show that FLOWING performs favorably compared to TPS warping.
>
> While dense optical flow is not directly applicable to the FRLL dataset, it raises an interesting direction for applying FLOWING to frame interpolation. We conducted a preliminary experiment on a single video with slow motion. Using RAFT [1] to compute dense optical flow, we trained an NCF model with sampled points displaced according to the flow. We then evaluated the quality of the reconstructed intermediate frame using MSE against the ground-truth frame:
>
> | Method | MSE↓ |
> | -------------------- | ---------------- |
> | Linear Blend    | 0.000264     |
> | Optical flow warp | 0.000049  |
> | NCF (Ours)      | **0.000040** |
>
> While preliminary, these results suggest that FLOWING is also competitive in optical-flow-based settings and may generalize to temporal interpolation tasks. Further experiments are underway to validate these findings.
>
> [1] Teed, Zachary, and Jia Deng. "Raft: Recurrent all-pairs field transforms for optical flow." European Conference on Computer Vision. Cham: Springer International Publishing, 2020.
>
> ## Questions
>
> > **How does FLOWING perform on datasets lacking clearly defined landmark structures, such as natural scenes with heavy occlusion or clutter, or on domains outside conventional faces (e.g., animals, medical images)? Is it possible to relax landmark reliance?**
>
> Our experiments in the paper do cover domains beyond conventional faces, including animals, clothing and general pictures; these can be found in the supplementary material. Nevertheless, relaxing this reliance is an important direction for future work. While FLOWING currently relies on landmarks for supervision, even generative methods such as DiffMorpher may struggle in cases where landmark structure is not exploited, as highlighted in Fig. 4 of the supplementary material.
>
> > **Could you quantify and compare compute/memory requirements and wall-clock times for the main approaches (FLOWING vs. ifmorph vs. classical), especially as input/image sizes or landmark counts scale?**
>
> Memory requirements for FLOWING and ifmorph training are not significantly affected by the number of landmarks, with usage around 2.5 GB for FLOWING and 1.5 GB for ifmorph. For compute requirements, we conducted experiments on an RTX 4090 GPU for both warp training and morphing (i.e., warping inference + blending). We report results for NODE trained with both 2k and 20k steps, as it converges quickly. Training times for NODE and ifmorph remain relatively stable across different landmark counts, unlike NCF, whose training time increases with more landmarks. Morphing time depends on image resolution: NODE offers faster training but slower morphing, while NCF has slower training and intermediate morphing speed.
>
> | Method | Landmark count | Training steps | Warp training time↓ |
> | ---------- | --------------------- | ------------------- | -------------------------- |
> | ifmorph | 68                      | 20k                  | 05m26s                    |
> | NCF (Ours) | 68                      | 20k                  | 08m48s                    |
> | NODE (Ours) | 68                      | 2k                    | 00m16s                    |
> | NODE (Ours) | 68                      | 20k                  | 02m39s                   |
>
> | Method | Landmark count | Training steps | Warp training time↓ |
> | ---------- | --------------------- | ------------------- | -------------------------- |
> | ifmorph | 130                      | 20k                  | 05m22s                    |
> | NCF (Ours) | 130                      | 20k                  | 15m05s                    |
> | NODE (Ours) | 130                      | 2k                    | 00m15s                    |
> | NODE (Ours) | 130                      | 20k                  | 02m38s                   |
>
> | Method | Image resolution | Morphing time (s)↓ |
> | ---------- | --------------------- | -------------------------- |
> |OpenCV | 256 x 256          | 0.01                   |
> | ifmorph | 256 x 256           | 0.05                    |
> | NCF (Ours) | 256 x 256    | 0.02                    |
> | NODE (Ours) | 256 x 256 | 0.03                   |
>
> | Method | Image resolution | Morphing time (s)↓ |
> | ---------- | --------------------- | -------------------------- |
> |OpenCV | 1350 x 1350          | 0.06                   |
> | ifmorph | 1350 x 1350           | 0.10                    |
> | NCF (Ours) | 1350 x 1350    | 0.31                    |
> | NODE (Ours) | 1350 x 1350 | 1.48                  |

---

> > ### Author Response · Authors · 2025-08-06
> >
> > Dear Reviewer Bay7,
> >
> > We appreciate the thorough review and constructive comments. We have submitted point-to-point responses to your questions and observations, including clarifications on the architecture's novelty and limitations, as well as additional experiments. We would appreciate it if you could inform us whether your concerns have been addressed. We are also happy to provide further clarification if needed.
> >
> > Best regards,
> >
> > Authors of paper #13820

---

> > > ### Comment · Reviewer_Bay7 · 2025-08-06
> > >
> > > Thanks for your thorough response.
> > >
> > > - **To our best knowledge, combining sinusoidal INRs and Neural ODEs has not been explored in previous research.**
> > >
> > > I recommend this publication, "Sun, Shanlin, et al. "Medical image registration via neural fields." Medical Image Analysis 97 (2024): 103249." Siren (sinusoidal INRs) + Neural ODE + Jacobian constraint has been applied. I am concerned about the novelty of the methodology.

---

> > > > ### Author Response · Authors · 2025-08-07
> > > >
> > > > Thank you for pointing out this reference, it appears we stand corrected. We will cite and discuss this work in the final version of the paper.
> > > >
> > > > Nevertheless, the referenced paper considers the problem of medical image registration, which is distinct from morphing. In that regard, we believe the other points raised in our rebuttal remain valid. As with prior works, we do not claim to be the first to apply flow-based architectures to imaging, though to our knowledge, we are among the first to explore Neural Conjugate Flows (NCFs) [1] for this purpose. Our main contribution lies in adapting these architectures to the domain of image and Gaussian splatting (3DGS) morphing. While image registration and morphing are related, morphing introduces several distinct challenges, including:
> > > > - Precise alignment of specific keypoints in feature-rich representations, as opposed to general shape matching;
> > > > - Smooth interpolation and blending between semantically distinct objects over time, with temporal consistency (see, e.g., the clothing items in the supplementary material). Note that 3DGS offers a novel of blending scheme between volumetric objects (Eq. 8);
> > > > - Handling large deformations and displacements (see, e.g., the animal examples in the supplementary material).
> > > >
> > > > In particular, morphing requires modeling coherent intermediate representations, whereas the referenced works use flow-based architectures solely to construct diffeomorphisms via time-one maps.
> > > >
> > > > Thus, we believe it is of potential interest to the morphing community that flow-based architectures can be successfully adapted to this setting. Indeed, our experiments show that FLOWING outperforms both classical baselines and modern INR-based alternatives, demonstrates strong results on morphing modern 3D representations (3DGS), and can be adapted to diffusion-based interpolation.
> > > >
> > > > [1] Bizzi et al. "Neural Conjugate Flows: A Physics-Informed Architecture with Flow Structure." Proceedings of the AAAI Conference on Artificial Intelligence. 2025

---

### Official Review · Reviewer_Qbyp · 2025-07-05

**Clarity:** 3
**Significance:** 3
**Originality:** 3
**Rating:** 5
**Confidence:** 4

**Summary:**

The paper proposes a structure preserving morphing mechanism which is based on Neural ODEs. The reason for using Neural ODE is because it naturally provides features like invertibility and temporarily smooth. This line of thinking has been used in several works such as in shape deformation so having a similar line o attack for the problem of morphing makes sense. The theoretical development presented in the paper is sound and rather straightforward. In addition to the flow based morphing approach, the paper leverages specialised implicit neural representations (INRs) to represent complex data such as images, etc. The paper includes several morphing experiments including extensions such as morphing of 3dGS and latent space morphing which open up exciting research directions.

**Questions:**

1. A section on limitations would be helpful.

2. I am not clear why a vanilla NODE applied on the features (where t=0 is source and t=1 is target) won't work for morphing? Can the authors elucidate this point in more detail?

**Ethical Concerns:**

["NO or VERY MINOR ethics concerns only"]

**Final Justification:**

I thank the authors for their response. I believe that my concerns are validated. I disagree with my fellow reviewers and find the paper contributions significant enough. Additionally I see that concerns regarding performance gains over prior methods are also substantial as noted in the paper as well as in the rebuttal. Thus, I vote to accept this paper.

**Limitations:**

yes

**Quality:**

3

**Strengths And Weaknesses:**

Strengths:

1. The simplicity of the framework, I argue is a major strength. NODEs are a very powerful class of models which come with several advantages such as invertibility by construction. Using them for the problem of morphing is an interesting idea.

2. Instead of naively applying NODE to morphing, the paper uses vision encoders to first map images in feature space and then learn the flows between source and target images. I think this underlying idea can very well be generalisable to other problems in computer vision and seeing its promise for the problem of morphing is interesting.

3. The paper shows both qualitatively and quantitatively superiority over prior art and includes interesting applications such as with 3DGS morphing and use of generative blending.


Weakness.

1. As with most NODE based approaches, I suspect there would be complications due to numerical errors in solving the initial value problem as well as slower speed. The paper hasn't sufficiently discussed these aspects.

---

> ### Author Rebuttal · Authors · 2025-07-30
>
> We thank the reviewer for the detailed comments. We are encouraged by the positive remarks regarding the simplicity of the framework, the use of flows for the problem of morphing as an interesting idea, the fact that the underlying idea can be generalizable to other problems, the qualitative and quantitative superiority over prior art, and interesting applications in 3DGS morphing / generative blending. We tried our best to address the reviewer’s concerns by performing additional experiments regarding numerical precision and computational requirements.
>
> ## Weakness
>
> > **1. As with most NODE based approaches, I suspect there would be complications due to numerical errors in solving the initial value problem as well as slower speed. The paper hasn't sufficiently discussed these aspects.**
>
> We use a fourth-order Runge-Kutta integrator with 10 steps, which should lead to errors on the order of $O(10^{-4})$.
> In practice, we observed no significant numerical error accumulation. In any case, this was one of the reasons why we included Neural Conjugate Flows as an integration-free alternative to Neural ODE. Additionally, we include an ablation study in which we compare inference trajectories with different number of steps against a 1000 steps baseline.  We perform this experiment using a FRLL sample (faces) and a Megadepth sample (monument views)
>
> FRLL sample
> | Steps | Max Squared Error |
> | ----- | ----------------- |
> | 2     | 5.12e-07          |
> | 5     | 6.19e-11          |
> | 10    | 6.37e-11          |
> | 20    | 6.19e-11          |
>
> Megadepth sample
> | Steps | Max Squared Error |
> | ----- | ----------------- |
> | 2     | 2.17e-05          |
> | 5     | 3.27e-09          |
> | 10    | 5.51e-10          |
> | 20    | 5.51e-10          |
>
> The results indicate that trajectories obtained using 10 and 1000 integration steps are nearly identical.
>
> We would also like to mention the difference between the number of integration steps during training and the number of integration steps during inference. The training integration steps are used to train a NODE with the desired properties. During inference, we need to integrate accurately enough to reflect the dynamics of the trained NODE model. Thus, the training integration steps and the inference integration steps do not need to coincide.
> We are happy to provide additional clarifications as needed, during the discussion period.
>
> ## Questions
> > **1. A section on limitations would be helpful.**
>
> We included a discussion of limitations in the supplementary material due to space constraints. For the final version, we will reorganize the manuscript to incorporate this section into the main paper, making it more visible to readers. Currently, one of the weaknesses of the proposed approach is its reliance on feature correspondences, as pointed out by other reviewers.
>
> > **2. I am not clear why a vanilla NODE applied on the features (where t=0 is source and t=1 is target) won't work for morphing? Can the authors elucidate this point in more detail?**
>
> We believe that “vanilla” in this context refers to NODEs without the sinusoidal activations and the Jacobian-based regularization. We do evaluate such NODEs, with results reported in Table 1 (NODE Sigmoid). Likewise, they are further explored in the ablation studies in the supplementary material (Section 2, Figure 1). These configurations yield inferior alignment and morphing quality, confirming the importance of our architectural and regularization choices. For the final paper, we will ensure that the distinction is clearer.

---

> > ### Author Response · Authors · 2025-08-06
> >
> > Dear Reviewer Qbyp,
> >
> > We appreciate the thorough review and constructive comments. We have submitted point-to-point responses to your questions and observations, including clarifications on FLOWING's performance as well as an additional experiment. We would appreciate it if you could inform us whether your concerns have been addressed. We are also happy to provide further clarification if needed.
> >
> > Best regards,
> >
> > Authors of paper #13820

---

### Note · Authors · 2025-08-11

We sincerely thank the reviewers for their thoughtful feedback and constructive suggestions. We appreciate their comments on the versatility and performance of the method and hope we have addressed their concerns by better detailing the framework’s novelty in the context of 2D/3D morphing and the choice of architectures used. We expect these comments to lead to an improved final paper.

Summary of key clarifications provided in the rebuttal:
- **Comparisons with Classical Methods** – Added results with thin-plate spline warping and optical-flow-based interpolation, showing that FLOWING outperforms both baselines.
- **Compute and Memory Analysis** – Provided detailed GPU time/memory tables for multiple resolutions, landmark counts, and training regimes.
- **OpenCV Baseline** – Justified its selection as a mature, widely used baseline; added challenging unaligned-face experiments showing that FLOWING offers strong results.
- **Scope and Novelty** – Clearly distinguished morphing from related problems (e.g., medical image registration, shape matching), emphasizing unique challenges such as time-consistent feature alignment, semantic blending, Gaussian splatting morphing, and large deformations; clarified the use of NCFs, as opposed to CNFs.
- **Novelty of Sinusoidal INRs + Neural ODEs** – Acknowledged the employment of these techniques in Sun et al. (2024), in the context of medical registration; highlighted that our work is the first to adapt NODEs and NCFs with sinusoidal INRs to morphing of images and 3DGS, addressing morphing-specific constraints and demonstrating strong results.

---

### Decision · Program_Chairs · 2025-09-17

**Decision:**

Accept (poster)

**Comment:**

This submission initially received mixed ratings: 2xBR and 2xA. On the positive side, reviewers appreciated effectiveness and efficiency of the proposed method, which promises to be generalizable and powerful. The experimental results back up these claims. At the same time, one of the reviewers raised concerns that a similar approach has been previously applied by Sun et al. (2024) in the domain of medical imaging. The other reviewer with a BR rating raised concerns about baselines, which were addressed in the rebuttal (but the rebuttal was not acknowledged by the reviewer and their final rating was not updated).
AC acknowledges certain similarities of the proposed method with Sun et al. (2024), but agrees with the authors that extended and a more generalizable framework with applications in generic image and 3D GS morphing will likely be of interest for the community. The final recommendation is to accept, with a strong recommendation to the authors to discuss this prior work in the camera-ready version of the paper.